# Heterologous Prime-Boost Vaccination with Commercial FMD Vaccines Elicits a Broader Immune Response than Homologous Prime-Boost Vaccination in Pigs

**DOI:** 10.3390/vaccines11030551

**Published:** 2023-02-25

**Authors:** Jaejo Kim, Seung-Heon Lee, Ha-Hyun Kim, Jong-Hyeon Park, Choi-Kyu Park

**Affiliations:** 1Animal and Plant Quarantine Agency, 177 Hyeoksin 8-ro, Gimcheon City 39660, Republic of Korea; 2College of Veterinary Medicine & Animal Disease Intervention Center, Kyungpook National University, Daegu 41566, Republic of Korea

**Keywords:** foot-and-mouth disease virus, cross-inoculation, heterologous prime-boost, homologous prime-boost, serological performance

## Abstract

Three commercial vaccines are administered in domestic livestock farms for routine vaccination to aid for foot-and-mouth disease (FMD) control in Korea. Each vaccine contains distinct combinations of inactivated serotype O and A FMD virus (FMDV) antigens: O/Manisa + O/3039 + A/Iraq formulated in a double oil emulsion (DOE), O/Primorsky + A/Zabaikalsky formulated in a DOE, and O/Campos + A/Cruzeiro + A/2001 formulated in a single oil emulsion. Despite the recommendation for a prime-boost vaccination with the same vaccine in fattening pigs, occasional cross-inoculation is inevitable for many reasons, such as lack of compliance with vaccination guidelines, erroneous application, or change in vaccine types by suppliers. Therefore, there have been concerns that a poor immune response could be induced by cross-inoculation due to a failure to boost the immune response. In the present study, it was demonstrated by virus neutralization and ELISA tests that cross-inoculation of pigs with three commercial FMD vaccines does not hamper the immune response against the primary vaccine strains and enhances broader cross-reactivity against heterologous vaccine antigens whether they were applied or not. Therefore, it could be concluded that the cross-inoculation of FMD vaccines can be used as a regimen to strategically overcome the limitation of the antigenic spectrum induced by the original regimen.

## 1. Introduction

Foot-and-mouth disease (FMD) is a contagious and economically important vesicular disease in cloven-hoofed animal species, including cattle, pigs, sheep, and goats [1]. The causative agent, foot-and-mouth disease virus (FMDV), belongs to the genus *Aphthovirus*, family *Picornaviridae*, and consists of seven serotypes: O, A, C, Asia 1, SAT 1, SAT 2, and SAT 3. Infection with one serotype does not confer cross-immunity to different serotypes [2,3]. Among the seven serotypes, type O and A FMDV are distributed globally, and the O type is the most endemic serotype [4]. Type O FMDV is divided into eight topotypes, CATHAY, Middle East–South Asia (ME-SA), Southeast Asia (SEA), Europe-South America (EURO-SA), Indonesia-1 and Indonesia-2 (ISA-1 and ISA-2, respectively), East Africa (EA), and West Africa (WA), based on 15% nucleotide differences [5]. Although type A FMDV is divided into three topotypes, ASIA, AFRICA, and EURO-SA, and it is considered to be antigenically and genetically the most diverse FMDV serotypes [6]. In addition, due to high antigenic diversity, the current vaccine, a purified inactivated virus vaccine derived from a specific strain can confer protective immunity against only a limited range of field strains [7].

Type O FMD outbreaks occurred twice in Korea between 2000 and 2002 and seven times between 2010 and 2019, and type A FMD outbreaks occurred three times between 2010 and 2018 [8,9,10,11,12]. After severe economic damage in 2010 and 2011, the Korean government implemented mandatory nationwide vaccination to control FMD [8]. Recently, three commercial vaccines were administered in domestic livestock farms. In the commercial vaccines, there are three distinct combinations of inactivated serotype O and A FMDV antigens: O/Manisa + O/3039 + A/Iraq formulated in a double oil emulsion (DOE) from Boehringer Ingelheim (BI, France), O/Primorsky + A/Zabaikalsky formulated in a DOE from the Federal Center for Animal Health (FGBI “ARRIAH”, Russia), and O/Campos + A/Cruzeiro + A/2001 formulated in a single oil emulsion (SOE) from Biogenesis Bago, Argentina. Despite the Korean government’s recommendation to prohibit cross-inoculation of vaccines, especially for prime-boost vaccination in fattening pigs, the occasional cross-inoculation of the vaccines on livestock farms is inevitable due to many reasons, such as a lack of compliance with vaccination guidelines, erroneous application, or change in vaccine types by suppliers. Therefore, there have been concerns that the level of the immune response expected with the prime-boost vaccination with the same vaccine cannot be reached due to failure of the boosting immune response if the booster vaccination is conducted with FMD vaccine that is different from the primary vaccine.

In this study, we cross-administered three commercial FMD vaccines to fattening pigs that had maternally derived antibodies (MDAs) from administration of FMD vaccines to sows, to determine whether cross-inoculation in the field might be one of the reasons for the low positive rates of some pig farms. The homologous and heterologous serological responses induced by the administration of the FMD vaccines were investigated by virus neutralization (VN) tests and enzyme-linked immunosorbent assays (ELISAs).

## 2. Materials and Methods

### 2.1. Cells, Viruses and Vaccines

Porcine kidney (LFBK) cells kindly provided by the Plum Island Animal Disease Center (New York, NY, USA) were used to culture FMDV. O_1_/Manisa/TUR/69 (O/Manisa), O/3039, and A_22_/Iraq/24/64 (A/Iraq) were received from BI. O/Primorsky/RUS/2014 (O/Primorsky, O/SEA/Mya-98 lineage) and A/Zabaikalsky/RUS/2013 (A/Zabaikalsky, A/ASIA/Sea-97 lineage) were received from ARRIAH. O_1_/Campos/BRA/58 (O/Campos), A_24_/Cruzeiro/BRA/55 (A/Cruzeiro), and A/ARG/2001 (A/2001) were provided by the FMDV WOAH reference laboratory at Sevicio Nacional de Sanidad y Calidad Agroalimentaria (SENASA, Argentina). Eight vaccine strains were cultivated in LFBK cells and prepared for VN tests.

Three bivalent commercial oil emulsion vaccines containing more than six 50% protective doses/dose, O/Manisa + O/3039 + A/Iraq, O/Primorsky + A/Zabaikalsky, and O/Campos + A/Cruzeiro + A/2001, were purchased from the corresponding manufacturers.

### 2.2. Genetic Comparison: Phylogenetic Analysis

The complete sequence of the VP1-coding region of FMDV from NCBI GenBank was used for phylogenetic analysis to determine the genetic distances among the original reference viruses of vaccine strains (O_1_/Manisa/TUR/69, O_1_/Campos/BRA/58, A_22_/IRQ/24/64, A/ARG/2001, and A_24_/Cruzeiro/Bra/55) and field isolates. Maximum likelihood trees were built using the program MEGA, version 11.0 [13], computing the evolutionary distances calculated using the Kimura-two parameter method and a bootstrap resampling analysis performed with 1000 replicates.

### 2.3. Vaccine Field Trial

A total of 198 crossbred pigs from a breeding farm in Chonnam Province were used for the vaccine field trial. All pigs were negative with FMD nonstructural protein (NS) antibodies using the PrioCHECK FMD nonstructural enzyme-linked immunosorbent assay (NS-ELISA) (Prionics AG, Schlieren-Zurich, Switzerland). Numbered ear tags were used to identify the individual pigs. Although all pigs were not vaccinated with FMD vaccines previously, they had MDA resulting from repeated administration of FMD vaccines in sows due to the mandatory vaccination policy. Therefore, to minimize statistically significant differences in MDA levels among fattening pigs, nine groups (*n* = 22 for each group) were allocated to have a relatively even distribution of the average and standard deviation among groups based on the results of VDPro FMDV type O Ab b-ELISA kit (Median Diagnostics, Korea) on sera at 7 weeks of age (Appendix A). Then, pigs in each group were vaccinated with each designated set of vaccine combinations for prime-boost vaccination at 8 and 12 weeks of age (Table 1). The nine vaccination strategies for the groups consisted of six heterologous booster regimens (A-B, A-C, B-A, B-C, C-A, and C-B) and three homologous booster regimens (A-A, B-B, and C-C). Blood samples were obtained at 7, 12, 16, 24, 28, and 32 weeks of age in every group to evaluate the serological response.

### 2.4. Serological Assays

The neutralizing antibody titers against FMDV in serum samples were measured as specified in the Manual of Diagnostic Tests and Vaccines for Terrestrial Animals of the World Animal Health Organization (WOAH Terrestrial Manual 2021) [3]. The VN titers were calculated as the log_10_ of the reciprocal serum dilution inhibiting cytopathic effects. The VN test was performed against all vaccine strains, and serum samples were also tested with a VDPro FMDV type O Ab b-ELISA kit, one of the recommended solid-phase competition ELISA (SPC-ELISA) test kits for serosurveillance in Korea, according to the manufacturer’s instructions to measure the antibody titers against type O structural protein (SP). In this study, 40% or higher inhibition (PI) was considered positive for type O SP by SP-ELISA. All serum samples obtained at 7 and 32 weeks of age were tested by a PrioCHECK FMDV NS Ab ELISA kit, according to the manufacturer’s instructions to detect antibodies against the nonstructrual 3ABC protein of FMDV, which is the highly conserved protein of the FMDV. Percent inhibition (PI)’ of 50% or greater was considered positive to NS antibodies. This NS-ELISA assay is one of the recommended assays to serologically differentiate the FMDV infected animals from animals vaccinated with FMD vaccines formulated with 146S antigen purified to reduce the NS protein content [3].

### 2.5. Analysis of Serological Relationships

According to the average VN titers at 16 weeks of age (4 weeks post-revaccination) in each homologous booster group (A-A, B-B, and C-C) in the field trial, the serological relationship of each vaccine strain with the homologous and heterologous vaccine strains was estimated in a similar format to the vaccine matching r_1_-value calculation. The r_1_-values were derived by dividing the reciprocal mean VN titers of sera against the homologous and heterologous strains by the reciprocal mean VN titers of sera against the vaccine antigen contained in the applied vaccine. The serological titers of sera corresponding to 1:64 or more against homologous strains were used for the calculation of the serological relationship. In the case of groups vaccinated with multiple same serotype strains, the higher homologous VN titer was used as the vaccine VN titer for the calculation of the r_1_-value.

### 2.6. Statistical Analysis

The results are expressed as the means ± standard errors of the means (SEMs). Because this field trial was designed to compare each serological result between the group boosted with the same vaccine as the primary applied vaccine and the group boosted with the different vaccine with primary applied one, the statistical analysis was conducted to determine the significant differences on serological effect between the homologous booster vaccinated group and heterologous booster vaccinated group within the same primed vaccinated groups. Therefore, the vaccine field trial data were analyzed using a mixed-effects model to assess the overall effects of the vaccine regimen on the serological level during the experiment. The mixed model defined the vaccine group, time, and interaction between vaccine group and time as fixed effects, whereas individual pigs were defined as a random effect. For the post hoc analysis, Tukey’s multiple comparison test was conducted to determine the significant differences at each collection day. All statistical analyses were performed using GraphPad Prism version 9.5.0 (GraphPad, San Diego, California, USA). *p* values < 0.05, < 0.01, < 0.001, or < 0.0001 were regarded as significant or highly significant.

## 3. Results

### 3.1. Phylogenetic Analysis

The results of the phylogenetic analysis of the VP1 gene sequence showed that the O/Manisa (AY593823.1), O/Campos (K01201.1), A/Iraq (AY593763.1), A/Cruzeiro (JQ082960.1) and A/2001 (JQ082983.1) strains were very genetically distinct. These strains were shown to be distant from Korean isolates. Korean isolates were distributed in the phylogenetic tree as clusters for three type O and one type A lineage: the three type O lineages were ME-SA/PanAsia, SEA/Mya-98, and ME-SA/Ind-2001 (Figure 1); the one type A lineage was ASIA/Sea-97 (Figure 2). Korean isolates were closely localized in the phylogenetic tree with those of other Southeast Asian countries, such as Nepal, China, Russia, and Vietnam.

### 3.2. VN Responses

VN titers against the type O vaccine strains in the vaccine field trial are summarized in Figure 3. In this field trial, most animals had preexisting VN titers against FMD type O vaccine strains. The average preexisting VN titer against the type O vaccine strains of all groups was more than 1.4 (log_10_). Most VN titers against type O vaccine strains were decreased at 12 weeks of age. These decreased VN titer were apparently observed against O/Manisa and O/Primorsky in the groups primed with O/Campos, C-C, C-A, and C-B. As the VN titers increased in most of graphs at 16 weeks of age (four weeks post-revaccination), most peak levels of VN titers were observed at 16 or 24 weeks of age. Although the VN titers increased in three homologous booster groups, A-A, B-B, and C-C, the increased VN titers in the heterologous booster groups tended to be comparable to or significantly higher than those in the homologous booster groups which were primed with the same FMD vaccine. These kinds of comparable or significant levels of VN titers against type O vaccine strains in heterologous booster vaccinated groups were observed until 32 weeks of age (20 weeks post-revaccination).

VN titers against the type A vaccine strains in the vaccine field trial are summarized in Figure 4. Depending on the type A vaccine strain, the average preexisting VN titers in the groups varied greatly. While the average preexisting titers against A/Iraq and A/Zabaikalsky were 1.78–2.04 and 1.48–1.80, the average preexisting titers against A/Cruzeiro and A/2001 were 1.10–1.57 and 0.95–1.36, respectively. After the primary vaccination at 8 weeks of age, the VN titer against type A vaccine strains in most groups was maintained or decreased at 4 weeks post-vaccination and increased at 4 weeks post-revaccination. After booster vaccination, the peak levels of VN titers were observed at 16 or 24 weeks of age. During the experiment, most type A VN titers in the heterologous booster groups were comparable to or significantly higher than those in the homologous booster groups within groups primed with the same vaccine. But exceptions were observed at a few time points, such as VN titer against the A/Cruzeiro at 24 and 28 weeks of age and the A/2001 at 28 weeks of age in groups vaccinated with A/Cruzeiro and A/2001. 

### 3.3. ELISA Responses

All sera collected at seven and 32 weeks of age were negative with FMD NS antibodies using the PrioCHECK FMD NS-ELISA. The highest percent inhibition was 41%. 

The ELISA PI and positive rate of the type O SP ELISA kit in the vaccine field trial are summarized in Figure 5 and Table 2, respectively. Similar to the VN test results, preexisting serological titers were detected in all groups. In the groups, the average PI and positive rate were more than 70.1% and 77.3%, respectively. Among the groups primed with same vaccines, the PIs of heterologous booster groups on each collection day were mostly comparable to those of the homologous booster groups. The significant high mean PI were observed in the heterologous booster groups among the groups primed with the O/Manisa + O/3039 + A/Iraq vaccine at 16, 24, and 38 weeks of age and the O/Primorsky + A/Zabaikalsky vaccine at 16 weeks of age. Within the homologous vaccine-primed groups, the positive rates of the heterologous booster groups on each collection day were mostly the same as or higher than those of the homologous booster groups (Table 2).

### 3.4. Serological Relationship between Vaccine Strains

The estimation of the serological relationship between vaccine strains is depicted in Table 3. Because two types of applied vaccines contained double type O or A vaccine strains, the higher titer for both homologous VN titers was considered the homologous vaccine VN titer. According to the estimation for type O vaccine strains, the r_1_-values of O/3039 with O/Campos and O/Primorsky were 0.69 and 0.25, respectively; the r_1_-values of O/Primorsky with O/Manisa, O/3039, and O/Campos were 0.43, 0.24, and 0.39, respectively; and the r_1_-values of O/Campos with O/Manisa, O/3039, and O/Primorsky were 0.61, 0.62, and 0.15, respectively. Regarding the results for type A vaccine strains, the r_1_-values of A/Iraq with A/Zabaikalsky, A/Cruzeiro, and A/2001 were 0.12, 0.23, and 0.01, respectively; the r_1_-values of A/Zabaikalsky with A/Iraq, A/Cruzeiro, and A/2001 were 0.68, 0.36, and 0.39, respectively; and the r_1_-values of A/Cruzeiro with A/Iraq and A/Zabaikalsky were 0.33 and 0.18, respectively.

## 4. Discussion

Considering that the O/3039 strain and O/Primorsky were reported to be very similar to O/TAI/Ban/60 (O/Cathay topotype) and O/Jincheon/SKR/2014 (O/SEA/Mya-98 lineage), respectively [11], and A/Zabaikalsky was reported to be very similar to A/Yeoncheon/SKR/2013 (O/ASIA/Sea-97 lineage) [14], all vaccine strains were genetically distinct from each other (Figure 1 and Figure 2). For this reason, some individuals have insisted that cross-inoculation in the field might be one of the reasons for the low positive rates of some pig farms in serological surveillance. In contrast to these concerns, inadequate poor immune responses induced by heterologous prime-boost vaccination were not observed in this present study. Rather than a failure of immune boosting caused by changing the booster vaccine, compared with the VN titers against the primed vaccine strains in homologous booster groups, comparable or significantly higher VN titers against the primed vaccine strains in heterologous booster groups were observed after boosting on each collection day in most application cases, with a few exceptions. In summary, broadened and enhanced cross-immunity by cross-inoculation with FMD vaccines in a prime-boost regimen in piglets was demonstrated in this study.

Even though each strain is genetically distinct, the serological relationship between the primed vaccine strain and the booster vaccine strain could be a more important factor influencing serological test results. In this study, we roughly estimated the serological relationship between primed and booster FMD strains using the average VN titers of each homologous booster group (groups A-A, B-B, and C-C) assayed at four weeks post-revaccination (16 weeks of age) (Table 3). The reasons for using VN titers at 16 weeks of age to estimate the r_1_-value in this study were that the average titers against homologous strains were the peak titers at this age in each homologous booster group (Figure 3 and Figure 4) and that the effect of MDA on the serological test was considered insignificant according to the near depletion of MDA after 12 weeks of age (98 days after birth) in other studies in pigs [15,16]. However, the results of r_1_-value estimation lacked consistency between vaccine strains, especially in the case using the average VN titers of the group that received the O/Primorsky and A/Zabaikalsky vaccines with other vaccine strains (Table 3). In groups A-A and C-C, A/Iraq was not matched with other vaccine strains, but the response showed slight cross-reactivity with A/Cruzeiro, according to the criteria of 0.3 for vaccine matching by the VN test [3]. However, this clear separation of the serological relationship was not shown by the r_1_-values of group B-B. According to other studies, the r_1_-value determined using higher titers of sera would be lower than that determined using lower titers of sera [17], and r_1_-values for the pool of pig sera tended to be higher than those for the cattle pool [18]. Therefore, this unclear distinction in the serological relationship in the r_1_-values of group B-B might be explained by the relatively large portion of low-titer porcine sera in group B-B. Therefore, we assumed that O/Primorsky was the most antigenically distinct among the type O vaccine strains, and A/Iraq might share some antigenicity with A/Cruzeiro but probably less antigenicity with A/Zabaikalsky or A/2001.

Similar to our study, no significant difference between calves vaccinated with the same or different vaccines as booster vaccines in VN titer against strains contained in the primed vaccine was observed in a cattle experiment [19]. Although the aim of the cattle study was to evaluate whether the interaction with different vaccines applied in prime-boost vaccination would interfere with protective efficacy, it was concluded that similar VN titers were to be expected because type O, A, and C vaccine strains contained in three different vaccines (Belgian, Dutch, and Italian trivalent vaccine) were antigenically closely related, except for the type A vaccine strain in the Dutch vaccine, according to the r_1_-values for the same serotype vaccine strains.

Studies of experimental swine H3N2 influenza virus inactivated vaccines [20,21] and avian H5N1 influenza virus inactivated vaccines [22] demonstrated that heterologous prime-boost vaccination with genetically and serologically distinct influenza vaccine antigens induced broader and more robust cross-reactivity with genetically and serologically distinct field isolates than homologous prime-boost vaccination, except for no observation of cross-reactivity with more distant subtype isolates. In the present study, there were limitations due to the use of commercial vaccines, for which we could not adjust the amounts of vaccine antigens and adjuvants, unlike the above studies of influenza vaccines, which included experiments with preparations containing the same amounts of antigens formulated with the same adjuvants. However, broader and more robust cross-reactivities with heterologous prime-boost vaccination mostly lasted for 24 weeks, although antigen concentrations and adjuvants vary in vaccines, which are usually confidential. We assumed that the features of different adjuvants in vaccines resulted in several significant decreases in VN titers with heterologous booster after 12 weeks post-revaccination, based on the specific serological patterns among groups.

For annual serosurveillance of FMD vaccine campaigns in Korea, type O SPC-ELISA kits were used for high-throughput and consistent performance, which were usually designed for detecting broad cross-reactivity against vaccine strains or field isolates [23]. The SPC-ELISA test is known to be less specific for types and strains than the VN test [24]. Therefore, it could be expected that comparable or highly significant serological reactivity in heterologous booster groups would be observed in the ELISA results, similar to the results depicted in Figure 5.

In the present study, it was demonstrated that cross-inoculation of pigs with three commercial FMD vaccines used in Korea did not hamper the immune response against the primary vaccine strains and enhances broader cross-reactivity against heterologous vaccine antigens regardless of whether they were applied. Therefore, it can be concluded that cross-inoculation of pigs with FMD vaccines can be a practical regimen to strategically overcome the limitation of the antigenic spectrum induced by prime-boost vaccination with one vaccine, considering that there is a possibility that future emerging FMDVs in Korea would be antigenically distinct from vaccine strains under intensive mass vaccination. However, further study might be needed to investigate the range of the antigenic spectrum induced by heterologous prime-boost vaccination. We believe that this is the first report demonstrating serological performance by cross-inoculation with different FMD commercial vaccines in pigs.

## Figures and Tables

**Figure 1 vaccines-11-00551-f001:**
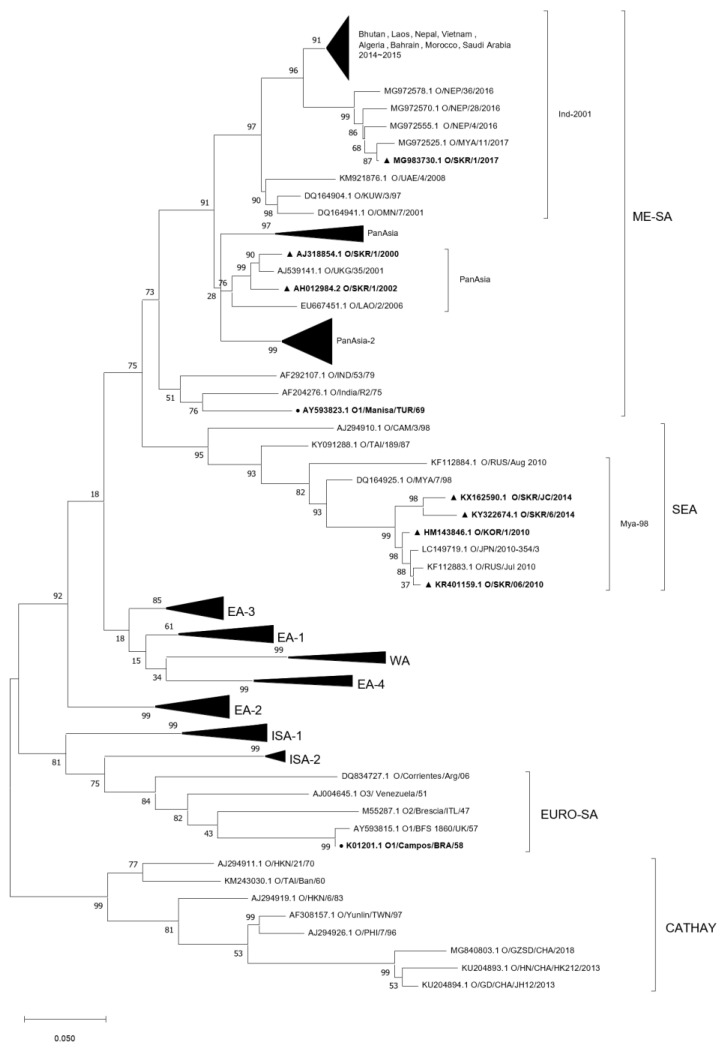
Phylogenetic tree showing the genetic relationships of FMDV type O vaccine strains and other isolates. Genetic distances were calculated with the Kimura two-parameter method based on the complete sequence of the VP1-coding region. A maximum likelihood tree was built using MEGA, version 11.0. The scale bar indicates nucleotide substitutions per site. Relevant lineages and topotypes are indicated with brackets. Black circles indicate the original reference viruses of vaccine strains. Black triangles indicate the Korean type O isolates since 2000.

**Figure 2 vaccines-11-00551-f002:**
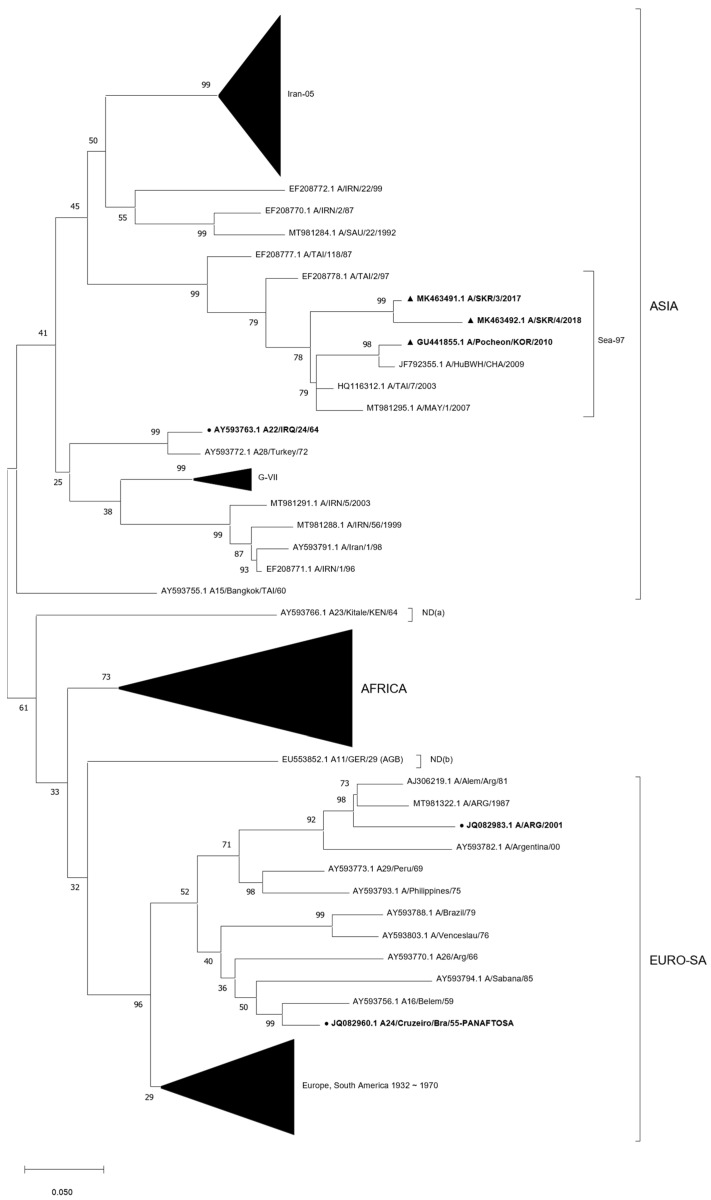
Phylogenetic tree showing the genetic relationships of FMDV type A vaccine strains and Korean isolates. Genetic distances were calculated with the Kimura two-parameter method based on the complete sequence of the VP1-coding region. A maximum likelihood tree was built using MEGA, version 11.0. The scale bar indicates nucleotide substitutions per site. Relevant lineages and topotypes are indicated with brackets. Black circles indicate the original reference viruses of type A vaccine strains used in Korea. Black triangles indicate the Korean type A isolates since 2010.

**Figure 3 vaccines-11-00551-f003:**
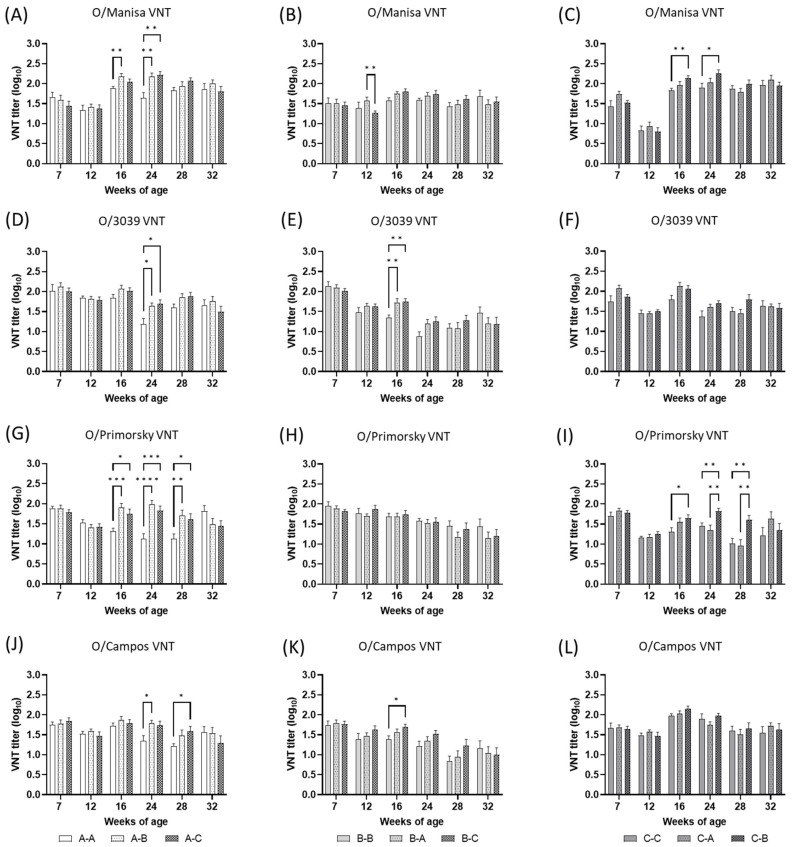
The type O FMDV-neutralizing antibody responses detected in serum samples from the farm-resident pigs of each group (*n* = 22), which were vaccinated with FMD vaccines. Blood was collected at 1 week before primary vaccination at 8 weeks of age and 4, 8, 16, 20, and 24 weeks after primary vaccination. Primary and booster vaccination were applied at 8 and 12 weeks of age, respectively. Nine groups were allocated according to the vaccine used: three homologous booster vaccination groups in which A-A, B-B, and C-C regimens were employed, and six heterologous booster vaccination groups in which A-B, A-C, B-A, B-C, C-A, and C-B regimens were employed. The results of the groups primed with O/Manisa + O/3039 + A/Iraq are depicted in (**A**,**D**,**G**,**J**). The results of the groups primed with O/Primorsky+A/Zabaikalsky are depicted in (**B**,**E**,**H**,**K**). The results of the groups primed with O/Campos + A/Cruzeiro + A/2001 are depicted in (**C**,**F**,**I**,**L**). The antigens used in the VN titer test are indicated at the top-center of each graph. Error bars represent the SEMs. * *p* < 0.05, ** *p* < 0.01, *** *p* < 0.001, **** *p* < 0.0001.

**Figure 4 vaccines-11-00551-f004:**
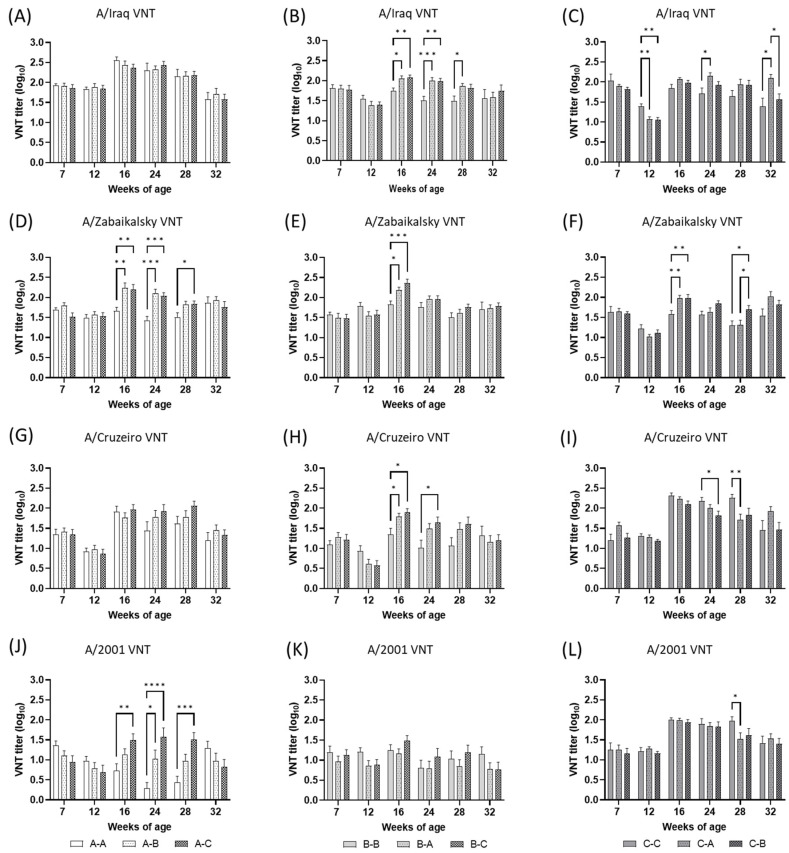
The type A FMDV-neutralizing antibody responses detected in serum samples from the farm-resident pigs of each group (*n* = 22), which were vaccinated with FMD vaccines. Blood was collected at 1 week before primary vaccination and 4, 8, 16, 20, and 24 weeks after primary vaccination. Primary and booster vaccination were applied at 8 and 12 weeks of age, respectively. Nine groups were allocated according to the vaccine used: three homologous booster vaccination groups in which A-A, B-B, and C-C regimens were applied, and six heterologous booster vaccination groups in which A-B, A-C, B-A, B-C, C-A, and C-B regimens were applied. The results of the groups primed with O/Manisa + O/3039 + A/Iraq are depicted in (**A**,**D**,**G**,**J**). The results of the groups primed with O/Primorsky+A/Zabaikalsky are depicted in (**B**,**E**,**H**,**K**). The results of the groups primed with O/Campos + A/Cruzeiro + A/2001 are depicted in (**C**,**F**,**I**,**L**). The antigens used in VN titer test are indicated at the top-center of each graph. Error bars represent the SEMs. * *p* < 0.05, ** *p* < 0.01, *** *p* < 0.001, **** *p* < 0.0001.

**Figure 5 vaccines-11-00551-f005:**
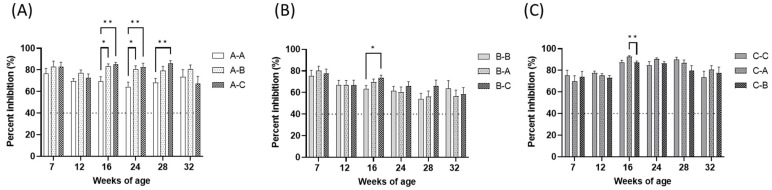
The ELISA responses detected in serum samples from the farm-resident pigs of each group (*n* = 22), which were vaccinated with FMD vaccines. Blood was collected at 1 week before primary vaccination and 4, 8, 16, 20, and 24 weeks after primary vaccination. Primary and booster vaccination were applied at 8 and 12 weeks of age, respectively. Nine groups were allocated according to the vaccine used: three homologous booster vaccination groups in which A-A, B-B, and C-C regimens were applied, and six heterologous booster vaccination groups in which A-B, A-C, B-A, B-C, C-A, and C-B regimens were applied. The results of the groups primed with O/Manisa + O/3039 + A/Iraq, O/Primorsky+A/Zabaikalsky, and O/Campos + A/Cruzeiro + A/2001 are depicted in (**A**–**C**), respectively. Error bars represent the SEMs. The horizontal dashed lines show the cutoff value of the ELISA, 40% percent inhibition (%). * *p* < 0.05, ** *p* < 0.01.

**Table 1 vaccines-11-00551-t001:** Experimental designs of the cross-inoculation regimens of three commercial FMD vaccines used in Korea.

Group	No. of Animals	Primary Vaccination (8 Weeks of Age)	Booster Vaccination (12 Weeks of Age)	Day Of Blood Collection (Weeks of Age)	Comments
		Vaccine Strain	Vaccine Strain		
A-A	22	O/Manisa + O/3039 + A/Iraq	O/Manisa + O/3039 + A/Iraq	7, 12, 16, 24, 28, 32	A homologous boost
A-B	22	O/Manisa + O/3039 + A/Iraq	O/Primorsky + A/Zabaikalsky	7, 12, 16, 24, 28, 32	A heterologous boost
A-C	22	O/Manisa + O/3039 + A/Iraq	O/Campos + A/Cruzeiro + A/2001	7, 12, 16, 24, 28, 32	A heterologous boost
B-B	22	O/Primorsky + A/Zabaikalsky	O/Primorsky + A/Zabaikalsky	7, 12, 16, 24, 28, 32	B homologous boost
B-A	22	O/Primorsky + A/Zabaikalsky	O/Manisa + O/3039 + A/Iraq	7, 12, 16, 24, 28, 32	B heterologous boost
B-C	22	O/Primorsky + A/Zabaikalsky	O/Campos + A/Cruzeiro + A/2001	7, 12, 16, 24, 28, 32	B heterologous boost
C-C	22	O/Campos + A/Cruzeiro + A/2001	O/Campos + A/Cruzeiro + A/2001	7, 12, 16, 24, 28, 32	C homologous boost
C-A	22	O/Campos + A/Cruzeiro + A/2001	O/Manisa + O/3039 + A/Iraq	7, 12, 16, 24, 28, 32	C heterologous boost
C-B	22	O/Campos + A/Cruzeiro + A/2001	O/Primorsky + A/Zabaikalsky	7, 12, 16, 24, 28, 32	C heterologous boost

**Table 2 vaccines-11-00551-t002:** Changes in the ELISA results in cross-inoculation trials with three commercial FMD vaccines.

Group	Weeks of Age at Blood Sampling (Weeks after Vaccination)					
	7 (−1 WPV ^1^)	12 (4 WPV)	16 (4 WPRV ^2^)	24 (12 WPRV)	28 (16 WPRV)	32 (20 WPRV)
A-A	20/22 ^3^ (90.9 ^4^)	22/22 (100)	20/22 (90.9)	19/22 (86.4)	20/22 (90.9)	20/22 (90.9)
A-B	20/22 (90.9)	22/22 (100)	22/22 (100)	22/22 (100)	21/22 (95.5)	21/22 (95.5)
A-C	20/22 (90.9)	21/22 (95.5)	22/22 (100)	21/22 (95.5)	22/22 (100)	18/22 (81.8)
B-B	20/22 (90.9)	18/22 (81.8)	21/22 (95.5)	20/22 (90.9)	13/22 (59.1)	14/22 (63.6)
B-A	21/22 (95.5)	19/22 (86.4)	21/22 (95.5)	18/22 (81.8)	16/22 (72.7)	16/22 (72.7)
B-C	21/22 (95.5)	16/20 (80.0)	20/20 (100)	19/20 (95)	16/20 (80.0)	14/20 (70.0)
C-C	20/22 (90.9)	22/22 (100)	22/22 (100)	21/22 (95.5)	22/22 (100)	18/22 (81.8)
C-A	17/22 (77.3)	22/22 (100)	22/22 (100)	22/22 (100)	21/22 (95.5)	22/22 (100)
C-B	19/22 (86.4)	22/22 (100)	22/22 (100)	22/22 (100)	21/22 (95.5)	21/22 (95.5)

^1^ Weeks post-vaccination. ^2^ Weeks post-revaccination. ^3^ No. positive animals/No. tested animals. ^4^ Positive rate of the serological results (%).

**Table 3 vaccines-11-00551-t003:** Average VN titers and the corresponding serological relationship (r_1_-value) of each vaccine against vaccine strains used in Korea.

Vaccine	*n*	Serotype	VNT Strain	VN Titer (log_10_)		r_1_-Value ^1^
				Mean	S.D.	
O/Manisa + O/3039 + A/Iraq	14	O	O/3039	2.05	0.27	1.00
			O/Manisa	2.02	0.08	0.93
			O/Primorsky	1.45	0.40	0.25
			O/Campos	1.89	0.23	0.69
	21	A	A/Iraq	2.60	0.32	1.00
			A/Zabaikalsky	1.70	0.39	0.12
			A/Cruzeiro	1.96	0.60	0.23
			A/2001	0.73	0.81	0.01
O/Primorsky + A/Zabaikalsky	7	O	O/Primorsky	2.11	0.38	1.00
			O/Manisa	1.74	0.30	0.43
			O/3039	1.48	0.36	0.24
			O/Campos	1.70	0.42	0.39
	10	A	A/Zabaikalsky	2.15	0.36	1.00
			A/Iraq	1.99	0.20	0.68
			A/Cruzeiro	1.71	0.27	0.36
			A/2001	1.74	0.34	0.39
O/Campos + A/Cruzeiro + A/2001	18	O	O/Campos	2.04	0.21	1.00
			O/Manisa	1.82	0.30	0.61
			O/3039	1.83	0.52	0.62
			O/Primorsky	1.21	0.56	0.15
	20	A	A/Cruzeiro	2.38	0.25	1.00
			A/Iraq	1.89	0.47	0.33
			A/Zabaikalsky	1.63	0.43	0.18
			A/2001	2.03	0.19	0.45

S.D. = standard deviation. ^1^ In the case of groups vaccinated with multiple same serotype strains, the higher homologous VN titer was used as the vaccine VN titer for the calculation of the r_1_-value.

## Data Availability

Not applicable.

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
