# Peer review of "Heterologous Prime-Boost Vaccination with Commercial FMD Vaccines Elicits a Broader Immune Response than Homologous Prime-Boost Vaccination in Pigs"

_vaccines, 2023, doi:10.3390/vaccines11030551_

Round 1
Reviewer 1 Report
Heterologous prime-boost vaccination with commercial FMD vaccines elicits a broader immune response than homologous prime-boost vaccination in pigs
Jaejo Kim, Seung-Heon Lee, Ha-Hyun Kim, Jong-Hyeon Park and Choi-Kyu Park
This is a relevant study however the presentation needs to be improved. It would have been better if more viruses were included see e.g., Tekleghiorghis et al 2014. For the statistical analysis of sequential data obtained on the same individuals a linear mixed model is more appropriate. The conclusions are probably similar but doing several ANOVAs on a large data set has the risk of cherry picking.
The objective for the study "In this study we determine whether the low antibody prevalence can be due to the fact that vaccines of different producers are used in the vaccination of pigs" was not clearly defined at the end of the introduction. The answer is clearly given in the results section.
I recommend major revision.
Tekleghiorghis, T., Weerdmeester, K., van Hemert-Kluitenberg, F., Moormann, R. J. M., Dekker, A. 2014. Comparison of Test Methodologies for Foot-and-Mouth Disease Virus Serotype A Vaccine Matching. Clinical and Vaccine Immunology 21(5); 674-683.
Comments:
Abstract:
Line 12: replace "have been" by "are"
Line 18 and 60: What do you mean with "lack of care" please describe more clearly
Introduction:
Line 31: Delete "highly". The reproduction ratio of FMD is high within a pen but not extremely high between pen or between farms.
Line 31: Replace "devastating" by "important"
Line 42: Genome changes are not equal to antigenic changes, and it is normal evolution. I doubt if the author of reference 6 is correct. I have never seen evidence that genomic evolution is faster in type A than in type O. The new type A outbreaks in the Middle East were introductions of other antigenic variants in Southeast Asia.
Materials and Methods:
Line 72: Did you use the LFBK cells expressing αVß6 bovine integrin? Please be specific. Change "Porcine kidney (LF-BK) cells" to "Porcine kidney cells expressing αVß6 bovine integrin (LFBK-αVß6)"
LaRocco, M., P. W. Krug, et al. (2013). "A Continuous Bovine Kidney Cell Line Constitutively Expressing Bovine αVβ6 Integrin Has Increased Susceptibility to Foot-and-Mouth Disease Virus." Journal of Clinical Microbiology 51(6): 1714-1720.
Line 81: The outcome of a PD50 experiment is the volume that protects 50% of the animals. The unit for the potency of a vaccine is therefore PD50/dose (not PD50 or 50% protective doses). Change "high-potency (more than six 50% protective doses)" to "containing >6 PD50/dose". The term "high-potency" is wrong, the WOAH standard is >3 PD50/dose, 2 times more is not high, but only slightly higher (in a potency test one cannot see the difference between 3 and 6 PD50/dose).
Line 94 – 98: It is not clear to me how the authors assigned the pigs to the groups. It is suggested that all groups had the same medium antibody titre for type A and O. But the criteria for inclusion are insufficiently clear described.
Line 111: Why were the titres dichotomized? What is the basis for selecting a 1:16 dose? Is this based on the 99% percentile in 1000 pigs without antibodies? The 99% percentile will be different for different strains.
Line 122 and elsewhere: Reclace "r value" by "r1-value". The r value is the pearson relation coefficient. You probably mean the r1-value which is the homologous part of the R = (r12 + r22)0.5. The r2-value is difficult to measure as we normally do not have 4 week post-vaccination sera directed against the field strain.
Line 127: The authors refer to reference 14 that states: "Pooling of serum samples significantly reduced the inter-animal and inter-trial variation". But by pooling serum samples one gets 1 result for each strain, and then no statistics can be performed. Of course, 1 titre has less variation than 5 titres from 5 animals, but the knowledge on the variability of the test is essential to do statistics. Testing the pooled sample repeatedly will give a more consistent result, but antigenicity seen by animals is different for each animal, therefore the variation in the animal population is relevant, not the variation in the test. So, this paper should not be referenced, it is wrong. Pooling samples before testing is wrong.
Line 131-137: In the results I see 6 dates per graph, 12 graphs and 2 serotypes which are compared by ANOVA. So, 144 comparisons. Was the Tukey correction using these 144 comparisons or only the comparison per graph (which is wrong). The only correct analysis method is a linear mixed effects model with the titre as result variable, the possible explanatory variables would be factor(age), vaccine group, VNT strain.
Results:
Line 168 – 169: Why dichotomize the VNT titres? How did you validate your cut-off?
Line 177: the abbreviations WPV and WPRV are not used 5 times in the main text. So, they should not be abbreviated. NB! In the tables the abbreviations are useful and explained correctly.
Figure 3 and 4 contain a red line that is not explained. It is not at the titre of 1.6 which is used to dichotomize the titres. It is unclear what this line means.
Line 213, 231 and 258: change "Positive rate of the serological result (%)" into "Percentage of pigs with a titre above 1.6"
Line 178 - 196: This is incomprehensible. Please help the reader in reading the graphs and identify the relevant issues. E.g. "In figure 3 the VNT titres measured against the O serotype strains are shown and in figure 4 the VNT titres measured against the A serotypes are shown. In the homologous vaccinated groups (A-B, B-B and C-C) the highest titres at week 16, 24, 28 and 32 were found in vaccine C-C for O/Manisa, vaccine A-A for O/3039, vaccine B-B for O/Primorsky and vaccine C-C for O/Campos. ……" Subsequently help the reader with what is shown on the mixed vaccine responses. Perhaps you can tell which group had the highest response against each VNT strain.
Line 237-243 Incomprehensible! I read "the heterologous booster group was comparable to the heterologous booster group", you probably mean something different. Start again with homologous vaccination groups, which one gives the highest response, then the heterologous vaccination groups.
Line 260-271: A table would probably be clearer than a graph in this section. The r1-values of O/Manisa and O/3039 are debatable as it was not a homologous O vaccine. In fact for all vaccines the r1-values are debate as the contained 2 serotypes.
The graph gives one value for the titre, but it should give the titres of all animals with SD of as box and whiskerplot. Antigenicity is measured because we want to know heterologous protection, and therefore the variation between animals is important in this analysis.
Discussion:
Line 283 - 293: is a repetition of the introduction. Delete this part. However, the objective is now clearer defined "cross-inoculation in the field might be one of the reasons for the low positive rates of some pig farms in serological surveillance". This should be reformulated as question and should be stated at the end of the introduction and the beginning of the discussion.
First start discussion your own results before mentioning results of others (line 309-317)
Author Response
Responses to reviewer #1:
We would like to thank the reviewer for careful and thorough reading of this manuscript and for the thoughtful comments and constructive suggestions, which help to improve the quality of this manuscript. Our response follows (the reviewer’s comments are in italics).
General Comments:
This is a relevant study however the presentation needs to be improved. It would have been better if more viruses were included see e.g., Tekleghiorghis et al 2014. For the statistical analysis of sequential data obtained on the same individuals a linear mixed model is more appropriate. The conclusions are probably similar but doing several ANOVAs on a large data set has the risk of cherry picking.
The objective for the study "In this study we determine whether the low antibody prevalence can be due to the fact that vaccines of different producers are used in the vaccination of pigs" was not clearly defined at the end of the introduction. The answer is clearly given in the results section.
I recommend major revision.
Reply:
We sincerely appreciate the positive and insightful feedback from the reviewer that help us upgrade the quality of this manuscript.
In response to the reviewer, we tried to provide the more precise and informative manuscript. But we are still sorry about some limitations of information.
As suggested by the reviewer, the data have been analyzed again with a mixed-effect model as described in line 142-156.
And the objective for the study has been added at the end of the introduction and redescribed at the beginning of the discussion.
Specific Comments:
Comments:
Abstract:
Line 12: replace "have been" by "are"
Reply
The correction has been made in line 12.
Line 18 and 60: What do you mean with "lack of care" please describe more clearly
Reply
As suggested by the reviewer, the description of “lack of care” were changed to “lack of compliance with vaccination guidelines” in line 18 and 59.
Introduction:
Line 31: Delete "highly". The reproduction ratio of FMD is high within a pen but not extremely high between pen or between farms.
Reply
The correction has been made in line 31.
Line 31: Replace "devastating" by "important"
Reply
The correction has been made in line 31.
Line 42: Genome changes are not equal to antigenic changes, and it is normal evolution. I doubt if the author of reference 6 is correct. I have never seen evidence that genomic evolution is faster in type A than in type O. The new type A outbreaks in the Middle East were introductions of other antigenic variants in Southeast Asia.
Reply
As suggested by the reviewer, the reference 6 was changed with the other reference. And the description about type A has been rewritten in line 40 to 42.
Materials and Methods:
Line 72: Did you use the LFBK cells expressing αVß6 bovine integrin? Please be specific. Change "Porcine kidney (LF-BK) cells" to "Porcine kidney cells expressing αVß6 bovine integrin (LFBK-αVß6)"
LaRocco, M., P. W. Krug, et al. (2013). "A Continuous Bovine Kidney Cell Line Constitutively Expressing Bovine αVβ6 Integrin Has Increased Susceptibility to Foot-and-Mouth Disease Virus." Journal of Clinical Microbiology 51(6): 1714-1720.
Reply
It is “LFBK cells”, not LFBK-αVß6. Although LFBK-αVß6 was also provided by Plum Island, the LFBK-αVß6 cell frequently used for isolation of field viruses.
Line 81: The outcome of a PD50 experiment is the volume that protects 50% of the animals. The unit for the potency of a vaccine is therefore PD50/dose (not PD50 or 50% protective doses). Change "high-potency (more than six 50% protective doses)" to "containing >6 PD50/dose". The term "high-potency" is wrong, the WOAH standard is >3 PD50/dose, 2 times more is not high, but only slightly higher (in a potency test one cannot see the difference between 3 and 6 PD50/dose).
Reply
As suggested by the reviewer, the statement has been rewritten in line 81 to 82.
Line 94 – 98: It is not clear to me how the authors assigned the pigs to the groups. It is suggested that all groups had the same medium antibody titre for type A and O. But the criteria for inclusion are insufficiently clear described.
Reply
To provide the clearer information, the statement has been rewritten in line 98-101. Because of MDA, we need pigs to be assigned to groups that have the relatively equal mean and standard deviation to minimize the difference among groups. But to sort the serological level of 198 animals, VNT was not practical to do this within one week, because VNT is the time consuming and labor-intensive test. So, we used the commercial type O SP-ELISA that can detect antibodies against type O FMDV broadly. The result is statistically insignificant ELISA levels among groups at 7 weeks of age depicted in Fig. 5 of this study
Line 111: Why were the titres dichotomized? What is the basis for selecting a 1:16 dose? Is this based on the 99% percentile in 1000 pigs without antibodies? The 99% percentile will be different for different strains.
Reply
The cut-off of the VNT titers was based on the criteria described in WOAH Terrestrial Manual. However, we decided not to describe the cut-off value of VNT in this study, because there is no problem explaining the VNT results, without the cut-off of VNT.
Therefore, as suggested by the reviewer, all data related with the cut-off of VNT have been deleted.
Line 122 and elsewhere: Reclace "r value" by "r1-value". The r value is the pearson relation coefficient. You probably mean the r1-value which is the homologous part of the R = (r12 + r22)0.5. The r2-value is difficult to measure as we normally do not have 4 week post-vaccination sera directed against the field strain.
Reply
As suggested by the reviewer, every description of “r value” has been rewritten as “r1-value”.
Line 127: The authors refer to reference 14 that states: "Pooling of serum samples significantly reduced the inter-animal and inter-trial variation". But by pooling serum samples one gets 1 result for each strain, and then no statistics can be performed. Of course, 1 titre has less variation than 5 titres from 5 animals, but the knowledge on the variability of the test is essential to do statistics. Testing the pooled sample repeatedly will give a more consistent result, but antigenicity seen by animals is different for each animal, therefore the variation in the animal population is relevant, not the variation in the test. So, this paper should not be referenced, it is wrong. Pooling samples before testing is wrong.
Reply
As suggested by the reviewer, the statement containing the word “pooling sample” has been deleted in the sentence in line 135-137.
Line 131-137: In the results I see 6 dates per graph, 12 graphs and 2 serotypes which are compared by ANOVA. So, 144 comparisons. Was the Tukey correction using these 144 comparisons or only the comparison per graph (which is wrong). The only correct analysis method is a linear mixed effects model with the titre as result variable, the possible explanatory variables would be factor(age), vaccine group, VNT strain.
Reply
As suggested by the reviewer, the data has been analyzed again.
Results:
Line 168 – 169: Why dichotomize the VNT titres? How did you validate your cut-off?
Reply
As suggested by the reviewer, all data related with the cut-off of VNT have been deleted.
Line 177: the abbreviations WPV and WPRV are not used 5 times in the main text. So, they should not be abbreviated. NB! In the tables the abbreviations are useful and explained correctly.
Reply
As suggested by the reviewer, WPV and WPRV have been rewritten as not abbreviated forms, but not in the table.
Figure 3 and 4 contain a red line that is not explained. It is not at the titre of 1.6 which is used to dichotomize the titres. It is unclear what this line means.
Reply
The dotted lines were the cut-off of VNT titer which is 1.2 log10, same as 1:16. But as suggested by the reviewer, all data related with the cut-off of VNT have been deleted.
Line 213, 231 and 258: change "Positive rate of the serological result (%)" into "Percentage of pigs with a titre above 1.6"
Reply
As suggested by the reviewer, all data related with the cut-off of VNT have been deleted.
Line 178 - 196: This is incomprehensible. Please help the reader in reading the graphs and identify the relevant issues. E.g. "In figure 3 the VNT titres measured against the O serotype strains are shown and in figure 4 the VNT titres measured against the A serotypes are shown. In the homologous vaccinated groups (A-B, B-B and C-C) the highest titres at week 16, 24, 28 and 32 were found in vaccine C-C for O/Manisa, vaccine A-A for O/3039, vaccine B-B for O/Primorsky and vaccine C-C for O/Campos. ……" Subsequently help the reader with what is shown on the mixed vaccine responses. Perhaps you can tell which group had the highest response against each VNT strain.
Reply
As suggested by the reviewer, the statement has been rewritten in line 184-210.
Line 237-243 Incomprehensible! I read "the heterologous booster group was comparable to the heterologous booster group", you probably mean something different. Start again with homologous vaccination groups, which one gives the highest response, then the heterologous vaccination groups.
Reply
As suggested by the reviewer, the statement has been rewritten in line 242-249.
Line 260-271: A table would probably be clearer than a graph in this section. The r1-values of O/Manisa and O/3039 are debatable as it was not a homologous O vaccine. In fact for all vaccines the r1-values are debate as the contained 2 serotypes.
The graph gives one value for the titre, but it should give the titres of all animals with SD of as box and whiskerplot. Antigenicity is measured because we want to know heterologous protection, and therefore the variation between animals is important in this analysis.
Reply
As suggested by the reviewer, the graph in fig. 6 has been changed to the table 3 containing means and SDs.
Discussion:
Line 283 - 293: is a repetition of the introduction. Delete this part. However, the objective is now clearer defined "cross-inoculation in the field might be one of the reasons for the low positive rates of some pig farms in serological surveillance". This should be reformulated as question and should be stated at the end of the introduction and the beginning of the discussion.
Reply
As suggested by the reviewer, the statement has been rewritten in line 285-299. The objective of this study has been added at the end of the introduction in line 64-67 and the beginning of the discussion in line 289-291.
First start discussion your own results before mentioning results of others (line 309-317)
Reply
As suggested by the reviewer, the paragraph discussing r1-value has been moved in line 300-323 before the results of others.
Reviewer 2 Report
Naive animals are essential for carrying out post-vaccination studies as any previous vaccination/infection will cause a secondary immune responds after vaccination.
In this manuscript non-naive animals were used. It is unclear what vaccination/exposure to FMD they had previously. Were all the animals vaccinated with the same vaccine? The authors say that these non-naive animals "were allocated to evenly distribute the significant differences" on day 0 sera; however, this isn't clearly explained nor is the data shown for the individual animals.
The suggestion would be to re-run the analysis using only naive animals as I am unsure how to comment or interpret any of the data. I have therefor not commented on the results and conclusion.
Minor comments:
Section 2.2 - certain vaccine viruses were not included in this analysis (such as O-3039), could this be explained? These viruses would have been passaged many times by the manufacturer which could cause changes in their genome. Would it therefor not be better to sequence the vaccine virus rather then relay on NCBI Gen Bank?
2.4 Serological assays - It is unclear what viruses were used for the VN test and why only type O was tested by ELISA.
In the abstract second line it should say foot-and-mouth not foot-and-mount
Author Response
Responses to reviewer #2:
We would like to thank the reviewer for careful and thorough reading of this manuscript and for the thoughtful comments and constructive suggestions, which help to improve the quality of this manuscript. Our response follows (the reviewer’s comments are in italics).
General Comments:
Naive animals are essential for carrying out post-vaccination studies as any previous vaccination/infection will cause a secondary immune responds after vaccination.
In this manuscript non-naive animals were used. It is unclear what vaccination/exposure to FMD they had previously. Were all the animals vaccinated with the same vaccine? The authors say that these non-naive animals "were allocated to evenly distribute the significant differences" on day 0 sera; however, this isn't clearly explained nor is the data shown for the individual animals.
The suggestion would be to re-run the analysis using only naive animals as I am unsure how to comment or interpret any of the data. I have therefor not commented on the results and conclusion.
Reply:
We appreciate your kind and insightful feedback.
In response to the reviewer, we tried to revise the manuscript to provide more precise and appropriate information. But we are still sorry about that information is still not enough to the reviewer.
As mentioned by the reviewer, naïve-animals must be used in this kind vaccination and effect study. Therefore, we used the unvaccinated and uninfected animals in this study. Because the Korean government conduct the mandatory massive FMD vaccination policy to sensitive animals, such as cattle, pigs and goats, sows need to be revaccinated at least twice a year according to FMD vaccine program. Therefore, sows have very high serological titers against FMDV due to the multiple FMD vaccination and these antibodies were delivered to their litters though colostrum. Because of this reason, studying with this much large size of sero-negative piglets is almost impossible. And even if such an experiment were conducted, the result would not be accepted because it is not same as the situation in Korea.
Because we used piglets having FMD antibodies, before vaccination experiment, we need pigs to be assigned to groups that have the relatively equal mean and standard deviation to minimize the difference among groups. But to sort the serological level of 198 animals, VNT was not practical to do this within one week, because VNT is the time consuming and labor-intensive test. So, we used the commercial type O SP-ELISA that can detect antibodies against type O FMDV broadly. The result is statistically insignificant ELISA levels among groups at 7 weeks of age depicted in Fig. 5 of this study.
And to confirm the infection status of animals, we used the NS-ELISA, which is generally used to detect FMDV-infected animals. All pigs were negative in the NS-ELISA.
We are really sorry for your inconvenience about not providing the clearer explanation about the animal status related with foot-and-mouth (FMD) vaccination and infection. For better understanding, we add the explanation related with animal status and test for infection in line 94-103 and 121-127.
Specific Comments:
Minor comments:
Section 2.2 - certain vaccine viruses were not included in this analysis (such as O-3039), could this be explained? These viruses would have been passaged many times by the manufacturer which could cause changes in their genome. Would it therefore not be better to sequence the vaccine virus rather than relay on NCBI Gen Bank?
Reply
As mentioned by the reviewer, the genetic information these viruses would be changed through several adaptations to harvest in cell culture system, which is vital for manufacturing commercial vaccines. However, when we were provided the vaccine viruses, we had to agree that the viruses are used for only VN testing. Any other implementation or study is not allowed, including release of the sequence data of the viruses, without permission.
And we think that genetic information in Fig. 1 and 2 and the information of other vaccine strains which are absent in NCBI Gen Bank in line 284-288 will be enough to describe the genetic distinction of vaccine viruses.
2.4 Serological assays - It is unclear what viruses were used for the VN test and why only type O was tested by ELISA.
Reply
Eight vaccine strains were used for the VN test. These are described at line 77-78 and line 115. And the captions of the figure 3 and 4 describe that legends at the top-center of each graph indicate the antigen used in the VN titer.
A Type A SP-ELISA was not used simply because a guaranteed commercial ELISA kit to detect type A FMDV with high sensitivity and specificity is not available so far. It is probably because type A FMDVs are genetically and antigenically extremely diverse.
In the abstract second line it should say foot-and-mouth not foot-and-mount
Reply
The correction has been made in line 13.
Round 2
Reviewer 2 Report
Thank you for explaining the reasons behind the study being designed using non-negative samples. This is now a lot more transparent and much clearer to the reader.
However, for complete transparency I would add a table of the individual pig results obtained at week 7. There is no transparency as to what individual animals went into the groups and what their VNT titres were to produce the average seen.
Figure 6 appears to be missing (line 305).
Minor Comments:
Table 1: this table would be viewed better in landscape as currently the columns are wrapping at strange places.
Line 142: the sentence starting with "because" should be re-written as it is unclear and an incomplete sentence.
Author Response
Responses to reviewer #2:
General Comments:
Thank you for explaining the reasons behind the study being designed using non-negative samples. This is now a lot more transparent and much clearer to the reader.
However, for complete transparency I would add a table of the individual pig results obtained at week 7. There is no transparency as to what individual animals went into the groups and what their VNT titres were to produce the average seen.
Figure 6 appears to be missing (line 305).
Reply:
As suggested by the reviewer, the individual ELISA data assayed at 7 weeks of age have been added as the supplementary table 1. In Korean pig farms, fattening pigs are housed together as groups according to the weight of individuals, because of feeding or husbandry reasons. Based on the group of the fattening pigs previously assigned, the number were randomly assigned by the farm. Because we could not move individuals to new assigned groups and we also had to minimize mistakes due to confusion when administering injections, we just conduct the minimal way to sort serological levels based on the results of type O ELISA. As you look in the supplementary table in line 434-437, the number 101-111 and 601-611, the number 112-122 and 612-622, the number 201-211 and 701-711, and the number 212-222 and 712-722 were allocated in each same group. And the other groups were allocated in the order assigned by the farm. After allocation, the vaccines were administrated as scheduled. For better understanding, the sentences in line 101-103 have been edited.
And figure 6 has been changed to the table format that is table 3, because it has been suggested by the other reviewer.
Minor Comments:
Table 1: this table would be viewed better in landscape as currently the columns are wrapping at strange places.
Reply
For better appearance, the contents of table 1 has been edited a little bit.
Line 142: the sentence starting with "because" should be re-written as it is unclear and an incomplete sentence.
Reply
The correction has been made in line 146.